# Impact of Somatic Vulnerability, Psychosocial Robustness and Injury-Related Factors on Fatigue following Traumatic Brain Injury—A Cross-Sectional Study

**DOI:** 10.3390/jcm11061733

**Published:** 2022-03-21

**Authors:** Daniel Løke, Nada Andelic, Eirik Helseth, Olav Vassend, Stein Andersson, Jennie L. Ponsford, Cathrine Tverdal, Cathrine Brunborg, Marianne Løvstad

**Affiliations:** 1Department of Research, Sunnaas Rehabilitation Hospital, Bjørnemyrveien 11, 1453 Nesoddtangen, Norway; marianne.lovstad@sunnaas.no; 2Department of Psychology, Faculty of Social Sciences, University of Oslo, 0316 Oslo, Norway; olav.vassend@psykologi.uio.no (O.V.); stein.andersson@psykologi.uio.no (S.A.); 3Department of Physical Medicine and Rehabilitation, Oslo University Hospital, Ullevål, 0424 Oslo, Norway; nadand@ous-hf.no; 4Institute of Health and Society, Center for Habilitation and Rehabilitation Models and Services (CHARM), University of Oslo, 0316 Oslo, Norway; 5Department of Neurosurgery, Oslo University Hospital, Ullevål, 0424 Oslo, Norway; ehelseth@ous-hf.no (E.H.); uxtvec@ous-hf.no (C.T.); 6Institute of Clinical Medicine, Faculty of Medicine, University of Oslo, 0316 Oslo, Norway; 7Psychosomatic and CL Psychiatry, Division of Mental Health and Addiction, Oslo University Hospital, 0424 Oslo, Norway; 8Turner Institute for Brain and Mental Health, School of Psychological Sciences, Monash University, Clayton, VIC 3800, Australia; jennie.ponsford@monash.edu; 9Monash-Epworth Rehabilitation Research Centre, Epworth Healthcare, Richmond, VIC 3121, Australia; 10Oslo Centre for Biostatistics and Epidemiology, Research Support Services, Oslo University Hospital, 0424 Oslo, Norway; uxbruc@ous-hf.no

**Keywords:** fatigue, rehabilitation, traumatic brain injury, neuropsychological function, PROMS

## Abstract

Fatigue is a common symptom after traumatic brain injuries (TBI) and a crucial target of rehabilitation. The subjective and multifactorial nature of fatigue necessitates a biopsychosocial approach in understanding the mechanisms involved in its development. The aim of this study is to provide a comprehensive exploration of factors relevant to identification and rehabilitation of fatigue following TBI. Ninety-six patients with TBI and confirmed intracranial injuries were assessed on average 200 days post-injury with regard to injury-related factors, several patient-reported outcome measures (PROMS) of fatigue, neuropsychological measures, and PROMS of implicated biopsychosocial mechanisms. Factor analytic approaches yielded three underlying factors, termed Psychosocial Robustness, Somatic Vulnerability and Injury Severity. All three dimensions were significantly associated with fatigue in multiple regression analyses and explained 44.2% of variance in fatigue. Post hoc analyses examined univariate contributions of the associations between the factors and fatigue to illuminate the relative contributions of each biopsychosocial variable. Implications for clinical practice and future research are discussed.

## 1. Introduction

Fatigue is a common symptom following traumatic brain injury (TBI) [1], with potentially severe impact on participation and quality of life [2], even when controlling for injury severity [3]. TBI is defined as “an alteration in brain function, or other evidence of brain pathology, caused by an external force” [4]. TBI is associated with increased mortality [5], and survivors may suffer from severe functional impairment, of which fatigue is often reported as a persistent problem in sub-acute and chronic phases following injury [6]. Fatigue is often defined as “an awareness of a decreased capacity for physical or mental activity, due to a perceived imbalance in the availability, utilization or restoration of energy that is needed to perform activities” [7]. A large number of heterogenous patient-reported outcome measures (PROMS) have been developed to evaluate subjectively experienced severity, characteristics and consequences of fatigue [8]. PROMS are, however, vulnerable to an assortment of potential biases [9], and there is currently no consensus for a single gold standard measure. A recent study evaluated the content overlap between items included in various fatigue PROMS often used in patients with stroke [10], showing that items from different PROMS may measure severity, characteristics, management or consequences of fatigue to varying degrees. Items from the Fatigue Severity Scale (FSS) [11], which is commonly used in patients with TBI, pertain primarily to the perceived consequences of fatigue. For a comprehensive measurement of fatigue, it is therefore necessary to expand the measurement using other PROMS and to establish whether fatigue can be construed as a unidimensional phenomenon across measures.

Conceptual models for the development and maintenance of fatigue after TBI and in other neurological disorders emphasize the heterogeneity in associated factors, spanning from premorbid characteristics, through primary injury-related factors, to secondary exacerbating factors [1,8]. The complex nature of fatigue and the abundance of implicated biopsychosocial factors necessitates an investigation of potential unifying mechanisms underlying the relationships between fatigue and associated constructs.

### 1.1. Mechanisms Associated with Fatigue

Demographic factors play an uncertain role in fatigue following TBI. Earlier studies have shown minimal or nonsignificant associations between fatigue, age and female gender [1,12,13,14], and a recent larger cohort study showed small but positive associations between fatigue, younger age, and female gender through the first six months post-injury [15]. This study further demonstrated an interaction between age and fatigue trajectory, with patients above 48 years of age reporting increasing, and younger patients decreasing, rates over the first 6 months. Of interest, injury severity does not seem to be consistently related to fatigue [1], with the caveat that most studies include a majority of patients with mild TBI. Cognitive deficits such as slowed information processing and attentional deficits have however been shown to be associated with increased levels of fatigue [16,17]. The coping hypothesis put forward by van Zomeren et al. [18] is one plausible explanation, in that cognitive deficits might result in increased energy expenditure during mental and physical exertion, which in turn may contribute to fatigue.

Beyond the direct effect of cognitive and other injury-related factors, an abundance of biopsychosocial mechanisms are implicated in onset and maintenance of fatigue. A conceptual model by Mollayeva et al. [1] emphasized the role of both TBI-specific as well as generic, non-injury-related mechanisms. A recent review [19] likewise established that there are several common risk factors for fatigue across neurological disorders, such as pre- and comorbid psychiatric symptoms, pain, sleep problems, and genetics.

Pain commonly co-occurs with fatigue after TBI [20,21] and is implicated as a central mechanism in fatigue across etiologies [22]. Beaulieu-Bonneau and Ouellet [23] found that pain was associated with fatigue 4 and 8 but not 12 months post TBI, indicating that this relationship may vary as an effect of time since injury.

Psychological distress (i.e., symptoms of depression and anxiety) is also related to fatigue following TBI [24,25,26,27,28]. While fatigue may by itself be a depressive symptom, fatigue may occur in isolation from depression in TBI and acquired brain injury [26], suggesting that the two are related, but distinguishable. Beaulieu-Bonneau and Ouellet [23] found depression to be associated with fatigue at 4, 8 and 12 months post-injury, indicating that these symptoms are intertwined over time. Symptoms of anxiety have also been linked with fatigue in isolation, although anxiety and depression frequently co-occur [27,29].

In addition to symptoms that may vary over time, people differ in their stable proneness for negative affect. Trait neuroticism as a five-factor personality trait has been extensively implicated as a possible precipitating mechanism in relation to fatigue in other populations, in epidemiological studies [30,31,32,33] and in mild TBI [34]. Merz et al. [34] also found negative associations between fatigue and trait agreeableness, conscientiousness and extraversion in patients with mild TBI. The role of neuroticism and other personality traits have, however, not been examined in relation to fatigue following more severe TBI. Trait optimism, furthermore, has been linked to better cognitive functioning after TBI [35], but has, to the best of our knowledge, not been examined in relation to post-TBI fatigue.

Daytime sleepiness and insomnia have been extensively studied in relation to fatigue following TBI [27,36,37]. For instance, Cantor et al. [14] demonstrated that fatigue and insomnia frequently co-occur, but that post-TBI fatigue may also occur without insomnia. Insomnia without post-TBI fatigue, however, was rare. As expected, daytime sleepiness was reported more frequently in patients with fatigue.

Motivational propensities for reward and punishment might additionally contribute to the development of fatigue. Behavioral inhibition (i.e., a tendency to be motivated by avoidance of unpleasant stimuli) and behavioral activation (i.e., a tendency to be motivated by the attainment of pleasure and reward) systems (BIS/BAS) were initially described by Carver and White [38]. A greater propensity for being motivated by avoidance of aversive stimuli and lower degree of reward responsiveness has been linked to fatigue in, e.g., multiple sclerosis [39]. The impact of BIS/BAS-propensities on fatigue has not, to the best of our knowledge, been examined in TBI.

Feelings of loneliness and isolation predict later development of both fatigue, pain and depression in non-TBI populations [40]. While loneliness has not been examined specifically as a risk factor for fatigue after TBI, loneliness is a common issue for people living with the chronic effects of TBI [41], leaving this factor of interest to explore.

Psychosocial resilience has been shown to predict increased participation following mild-severe TBI [42], and to predict longitudinal decreases in fatigue following mild TBI [43] but has not been studied extensively with regard to post-TBI fatigue.

### 1.2. Clinical Complexity

In summary, fatigue following TBI has a demonstrable impact on quality of life and functional recovery, and an abundance of mechanisms could potentially be implicated in the precipitation, initiation and maintenance of fatigue following TBI. The factors involved may act in isolation, their effects may be summed, and they may interact with each other in dynamic ways. An obstacle in studies involving vulnerability and protective factors is that inferences drawn from models incorporating only a few factors may not provide a comprehensive understanding of possible underlying constructs. A clearer picture of the underlying clustering of vulnerability and protective factors, however, may inform further research in selection of the most essential constructs in fatigue models, and inform clinical decision making.

### 1.3. Study Aims

The primary aim of this study was to enhance our theoretical understanding of the relationship between fatigue and injury-related, cognitive and self-reported biopsychosocial factors. A factor analytic approach was used to (1) examine if fatigue could be construed as one single outcome across several measures, and (2) examine potential underlying dimensionality of several injury-related, cognitive and psychosocial measures commonly associated with fatigue. Finally, we aimed to (3) explore the relevance of these dimensions to fatigue 6 months after TBI.

## 2. Materials and Methods

### 2.1. Recruitment

The study includes the first wave from a prospective observational study of patients with TBI conducted from 2018–2021. Included patients were injured between January 2018 and April 2020 and admitted to the Neurosurgery department at Oslo University Hospital (OUH). OUH is the only Level I trauma center with neurosurgical services in the south-eastern region of Norway with a population base of more than half of the Norwegian population (i.e., 2.9 million).

Injury characteristics and clinical data from the acute hospital stay were retrieved from the Oslo TBI Registry—Neurosurgery, a quality database at OUH [44]. The remaining variables were measured approximately 6 months post-injury. Inclusion criteria were patients between 18–65 years of age, admitted with TBI (ICD-10 diagnoses S06.1–S06.9), herein defined as patients presenting with intracranial injury (as confirmed by computed tomography (CT) or magnetic resonance imaging (MRI)) during the acute phase, and who have survived until six months post-injury. Exclusion criteria were pre- and comorbid diagnoses of severe mental illness or neurological disorders, ongoing substance or alcohol abuse, non-fluency in Norwegian or English, and severe functional impairment hindering completion of the study protocol (i.e., disorders of consciousness, persistent severe anosognosia and severe motor deficits). Patients were identified prospectively after admission to the Neurosurgical department at OUH. Patients were recruited through clinical follow-up consultations at Sunnaas Rehabilitation Hospital and the Department of Physical Medicine and Rehabilitation at OUH. Patients not followed up at these institutions received an invitation to participate by mail.

### 2.2. Injury Characteristics

Pre-injury physical health status was scored using the American Society of Anesthesiologists’ physical status classification (ASA-PS), with scores ranging from 1 to 6 depending on the absence or presence of various severities of systemic disease premorbid to injury [45], with increasing scores indicating more severe disease.

Several indicators of injury severity were included. Lowest Glasgow Coma Scale (GSC) score ranged from 3–15 registered at injury site, or admission to hospital pre-intubation was registered, as well as GCS upon discharge from the acute hospital. Rotterdam CT score is a prognostic classification of traumatic brain injuries scored on the basis of grade of compression of the basal cisterns, the presence of a midline shift, epidural mass lesion, and intraventricular blood or tSAH [46], with higher scores indicating more severe injuries. The Head Abbreviated Injury Scale (AIS_head) version 1998 [47] was used to describe the anatomical severity of injury. AIS classifies injuries to various body regions ranging from minor (1) to fatal (6). We dichotomized AIS_head scores into AIS < 4 (less severe) and AIS ≥ 4 (very severe injury) for descriptive analyses but used the ordinal scale scores in subsequent analyses. Finally, discharge destination from the acute hospital was registered. For this study, a dichotomous dummy variable was generated for those who were referred through a direct pathway into rehabilitation units.

### 2.3. Measures

#### 2.3.1. Fatigue

The Fatigue Severity Scale (FSS) [11] contains 9 items and asks the participants to rate the degree of interference from fatigue in various functional domains on a Likert scale from 1 to 7, with higher scores indicating higher degree of fatigue interference. Norwegian norms adjusted for age, gender and education are available [48]. The FSS has good psychometric qualities [48].

Chalder Fatigue Scale (CFQ) [49], has been applied primarily in research into chronic fatigue syndrome (CFS) and myalgic encephalomyelitis (ME), but also in neurological populations such as stroke [50]. Patients are asked to rate 11 items pertaining to physical and cognitive/mental symptoms of fatigue within the last month. The CFQ uses a four-point response scale where 0 = “less than usual”, 1 = “no more than usual”, 2 = “more than usual” and 3 = “much more than usual”. Normative data from the general population exist, grouped by age and gender [51].

The fatigue subscale of Giessen Subjective Complaints List (GSCL) [52] has been used within psychosomatic and epidemiological studies. The fatigue subscale includes 6 items, rating the presence of fatigue symptoms in general on a five-point scale from 0 = “not at all” to 4 = “strongly”.

Finally, one item from the Rivermead Post-Concussion Symptoms Questionnaire (RPQ) [53] asks the participants to rate the presence of fatigue on a scale from 0 to 4, where 0 = “not a problem”, 1 = “no longer a problem”, 2 = “a mild problem”, 3 = “a moderate problem”, and 4 = “a severe problem”. This single item is often used to assess fatigue in patients with concussion and TBI in clinical settings, and a recent multicenter TBI study employed it as a primary outcome measure of fatigue [15].

#### 2.3.2. Neuropsychological Tests

Cognitive functioning was assessed with the following neuropsychological measures:

The Matrix Reasoning and Similarities subtests from Wechsler’s Abbreviated Scale of Intelligence (WASI) [54] were included as measures of abstract reasoning abilities. Auditory attention and working memory were assessed with Digit Span from Wechsler’s Adult Intelligence Scale IV (WAIS-IV) [55]. Psychomotor speed was assessed with Trail Making Test (TMT) subtests 2–3 and Color-Word Interference Test (CWIT) subtests 1–2 from Delis–Kaplan Executive Function System (D-KEFS) [56]. Subtest 4 from the TMT and subtests 3–4 from the CWIT furthermore provide measures of executive function/mental flexibility. The Conners Continuous Performance Test III (CPT-III) [57] was included as a measure of sustained and focused attention. The change in coefficient of variation (CoV), a measure of increase in intraindividual variability in reaction times from the first to the second half of the test, was computed. CoV is calculated by dividing the standard deviation of reaction times (RT) by the average RT within the individual [58], and the measure of change in CoV was calculated by subtracting the CoV for the first three blocks from the last three blocks (CoV block change).

#### 2.3.3. Secondary PROMS

Psychological distress over the last two weeks was measured using a 10-item short version of Hopkins Symptom Checklist [59,60], with subscales for (1) depressive and (2) anxiety symptoms.

Five-factor personality traits were measured using the NEO Five Factor Inventory 3 (NEO-FFI-3) [61], which provides gender-corrected normative scores on trait neuroticism, conscientiousness, extroversion, agreeableness and openness to experience. The inventory contains 60 items, with 12 items pertaining to each personality trait.

Behavioral inhibition and activation tendencies were measured using The Behavioral Inhibition System/Behavioral Activation System (BIS/BAS) Scale [38], which contains one subscale for BIS, and three subscales for the BAS, namely (1) reward responsiveness, (2) drive, and (3) fun seeking.

Loneliness was measured using three items from the UCLA Loneliness Scale, Version 3 [62].

Trait optimism was measured with six items from the optimism subscale of the Life Orientation Test Revised (LOT-R) [63].

Resilience was measured with the Resilience Scale for Adults (RSA) [64], with subscales for facets of resilience, namely (1) planned future, (2) social competence, (3) family cohesion, (4) perception of self, (5) social resources, and (6) structured style.

Somatic symptom burden was assessed with subscales from Giessen Subjective Complaints List (GSCL) [52], regarding the presence of (1) gastrointestinal symptoms, (2) musculoskeletal symptoms, and (3) cardiovascular symptoms. Pain localization was assessed using a pain drawing [65], with higher scores indicating generalized pain dispersed across several bodily regions. Pain severity across the last two weeks was assessed with Numerical Rating Scales (0–10, where 10 indicates most severe pain) [66], asking the participants to rate (1) the lowest pain severity, (2) the highest pain severity, (3) the average pain severity, and (4) the current pain severity.

Daytime sleepiness was measured with the Epworth Sleepiness Scale [67], which asks respondents to rate the probability of falling asleep throughout a range of daily activities. Subjective sleep deficits were measured with the Insomnia Severity Index [68], which rates the presence of difficulties with falling asleep, staying asleep, early awakening, and the functional impact of sleep problems.

#### 2.3.4. Functional Outcome

Global functional impairment upon discharge from the acute hospital stay was estimated with the five-level Glasgow Outcome Scale (GOS) [69], while functional outcome 6 months post-injury was assessed with the eight-level Glasgow Outcome Scale Extended (GOSE) [70], which categorizes patients based on their degree of return to work, vocational and leisure activities, social and emotional symptoms and a variety of other persistent complaints following injury. Lower scores indicate greater functional impairment.

### 2.4. Analyses

All analyses were conducted in SPSS, version 27 [71]. Preliminary Pearson correlation analyses were conducted to evaluate bivariate relations between the various measures of fatigue, sociodemographic variables, injury-related factors, neuropsychological measures and self-reported psychosocial constructs.

#### 2.4.1. Dimension Reduction

In order to ascertain a fatigue factor possibly reflecting a unidimensional phenomenon in our TBI sample, a factor analysis was conducted on FSS, CFQ, the fatigue subscale from GSCL, and the fatigue item from RPQ. Items pertaining specifically to cognitive complaints (CFQ items 8–11 and GSCL item 15) and daytime sleepiness (CFQ item 3 and GSCL item 4 and 14) were excluded from these analyses to avoid item overlap between fatigue and independent variables.

Furthermore, an exploratory factor analysis was conducted on all variables (PROMS, neuropsychological and injury-related) with significant (*p* < 0.05) bivariate associations with either one or several of the fatigue measures. Due to the exploratory aim of the study, variables approaching significance (i.e., *p* < 0.08) were also included. Factors with eigenvalues above 1 were first generated in line with the Kaiser Guttman criterium. A scree plot was generated and inspected according to Cattell’s criterium [72]. Parallel analyses were performed to generate significant eigenvalues for factor retention [73], which has been shown to be a more consistently accurate method for factor retention decisions [74]. Oblimin oblique rotation was conducted to allow factors to correlate. Saliency of factor loadings was evaluated for significance (*p* < 0.05) according to the formula proposed by Norman and Streiner (2014), providing a cut-off for salient loadings at 0.40. Variables not loading significantly on any of the factors were removed, and the analyses were repeated without them. In the case of cross-loading variables, variables were selected on the basis of the strength of their loadings, as well as their conceptual alignment with the factor on the whole. New factor analyses were then conducted for each factor, including only those variables saliently loading on the factor. Factor scores were generated through regression.

Factor reliability was assessed for all resulting factors, through the calculation of Cronbach’s alpha with standardized variables, with negatively loading variables reversed. Alpha values of 0.70 or higher were deemed acceptable, and values of 0.90 or higher were considered excellent.

#### 2.4.2. Multiple Regression

In order to evaluate the relations between fatigue and the factors derived from the previous step, the fatigue factor was regressed on the factor scores from associated constructs. Variables were entered into the linear regression model blockwise. Sociodemographic variables were entered first, with age (centered around the sample mean of 45), educational attainment (centered around the sample mean of 13 years), and gender (female) as baseline covariates. The factors from the previous step were then added to examine if they contributed significantly to the model. Changes in F-scores were evaluated for significance in model improvement across each block. Bootstrapping was conducted to evaluate the robustness of the regression coefficients, and a 95% confidence interval (CI) was produced based on 2000 random draws from the sample. The results from linear regression analyses are reported with unstandardized regression coefficients (B) with bootstrapped standard errors (SE), 95% confidence intervals (CI), standardized regression coefficients (β) and explained variance (adjusted R^2^).

Partial regression plots were generated to evaluate the impact of potential outliers. Residual plots were also inspected to evaluate deviance from assumptions of normality, homoscedasticity and linearity. Residual scores were finally checked for associations with variables not included in previous factor analyses, to evaluate potential residual effects not captured by this model. Post hoc analyses were then conducted to evaluate the potential additional explanatory value of these variables. Finally, univariate regression analyses were conducted post hoc to evaluate the associations between individual variables contained within each factor, and the fatigue factor.

## 3. Results

### 3.1. Sample Characteristics

A total of 96 patients were included. See Figure 1 for an overview of the exclusion and inclusion process.

The average age was 45.3 (SD = 13.9), with a mean educational attainment of 13.5 years (SD = 2.3). The sample consisted of 77 (80.2%) males and 19 (19.8%) females.

On the ASA-PS, 69 patients (71.9%) were classified as healthy prior to injury, 19 (19.8%) as having moderate organic disease not impairing function, and eight patients (8.3%) as having severe organic disease.

The sample mean of GCS registered at injury site or at admission to the hospital pre-intubation was 10.7 (SD = 3.6), while GCS registered upon discharge from the acute hospital was 14.4 (SD = 0.9). The sample mean Rotterdam CT score was 2.8 (SD = 0.9). Using the dichotomized AIS_head classification, 18 patients (18.8%) were classified within the less severe category, and 78 (81.3%) within the very severe category. Upon discharge from the acute hospital, GOS ratings based on medical records classified 39 patients (40.6%) with moderate disability, 56 (58.3%) with severe disability, and one patient (1.1%) as being in a vegetative state.

Fifteen patients (15.6%) were discharged directly to their homes, 32 (33.3%) to a local hospital, and 49 (51%) were referred to a rehabilitation unit.

The study assessment was conducted on average 205 days (SD = 28) since injury.

### 3.2. Fatigue PROMS

The FSS demonstrated good internal consistency (α = 0.91). The average score was 3.7, corresponding to a demographically corrected T-score of 48.8 (SD = 11.9).

CFQ demonstrated good internal consistency (α = 0.89). The mean sum score for the total scale was 16.2, corresponding to a demographically corrected T-score of 60.8 (SD = 14.2), with comparable results on the mental/cognitive and physical subscales. Items 1 and 2 on the CFQ ask the patients to rate whether they experience increased tiredness or an increased need for rest within the last month compared to their habitual function, and 58 (60.4%) and 59 (61.5%) patients, respectively, endorsed the presence of these problems as compared to their habitual function.

The GSCL subscale demonstrated good internal consistency (α = 0.89). On the GSCL fatigue subscale, the mean score was approximately 1 (SD = 0.9), corresponding to the response category “somewhat a problem”.

On the RPQ fatigue item, 47 patients (49%) reported at least mild problems with fatigue, and 27 (28.1%) reported moderate-severe problems. For an overview over bivariate correlations between fatigue PROMS, see the Appendix A.

### 3.3. Fatigue and Associated Factors

Overall, fatigue as measured with several PROMS was consistently associated with several biopsychosocial PROMS and functional outcome, while the associative patterns were less consistent for injury-related and neuropsychological variables. There were no bivariate associations between demographic variables (age, gender, education) and any of the fatigue measures. The Head Abbreviated Injury Scale (AIS_head), length of acute hospital stay, GOS at discharge from acute hospital, and having a direct pathway to rehabilitation were associated with higher Physical Fatigue on CFQ. GCS at discharge trended toward significance (*p* < 0.08) in its relationship with the Physical Fatigue subscale from CFQ. No other measure of fatigue was significantly associated with variables from the acute phase.

Fatigue scores (FSS, CFQ, GSCL and RPQ) were positively associated with depression, anxiety, trait neuroticism, daytime sleepiness, insomnia, behavioral inhibition (BIS), all measures of pain, loneliness, and somatic (musculoskeletal/gastrointestinal/cardiovascular) symptom burden, albeit with some variation across measures. Trait openness was positively associated with the RPQ fatigue item only.

Fatigue was negatively associated with two resilience subscales (perception of self and planned future) on most fatigue PROMS, and trending toward significance (*p* < 0.08) for trait optimism in association with the FSS. Trait extraversion was negatively associated with the GSCL fatigue subscale only, and trait conscientiousness was trending toward significance (*p* < 0.08) for a negative association with the FSS.

Fatigue was negatively associated with performance on the CWIT 4—Switching Condition (a measure of mental flexibility) for the FSS and CFQ, and FSS was negatively associated with performance on measure of intraindividual stability of sustained reaction times on the CPT-III. The mental fatigue subscale on the CFQ was negatively associated with performance on several neuropsychological measures. However, this subscale probes about subjective cognitive complaints such as memory and word-finding difficulties, and these associations are not taken into account in the following analyses.

All measures of fatigue were negatively associated with functional outcome 6 months post-injury as measured by GOSE.

For a complete overview of bivariate associations between fatigue and included variables, see the Appendix A.

### 3.4. Dimension Reduction

In the factor analysis of the items from the included fatigue outcome measures, three factors were initially generated with an eigenvalue above 1. Both an inspection of the scree plot and parallel analysis of critical threshold for significant eigenvalues provided support for a one-component solution. Items 1 and 2 from the FSS were excluded following the primary factor analysis due to non-salient loadings on the generated factor. All remaining items loaded saliently on the single component (see Table 1). The factor demonstrated excellent reliability (Cronbach’s alpha = 0.95) and thus provided an opportunity to examine relationships between the other variables and one single and robust fatigue measure.

For the factor analysis of all associated constructs, seven components were initially generated with an eigenvalue above 1. While the inspection of the scree plot of eigenvalues might suggest retention of either three or four components according to Cattell’s criterium, the thresholds from the parallel analysis supported the retention of only the first three components. The component matrix was obliquely rotated using Oblimin rotation, which allows for correlated components. The neuropsychological measures (CWIT-4 and CPT-III CoV Block Change) and trait openness did not load saliently on any of the three factors, and the analysis was repeated without these variables included.

Based on the salient positive loadings from resilience subscales, trait optimism, trait extraversion and trait conscientiousness on Factor 1, this component was designated as a Psychosocial Robustness factor. Factor 1 also has salient negative loadings from trait neuroticism, behavioral inhibition, symptoms of depression and anxiety, loneliness, and gastrointestinal and cardiovascular symptoms, confirming that robustness is a combination of presence of positive protective factors, but is also an absence of risk factors. Factor 2 had salient loadings from all measures of pain, somatic symptom burden (musculoskeletal, cardiovascular and gastrointestinal), daytime sleepiness, subjective sleep complaints, as well as symptoms of depression and anxiety. This factor was thus designated as a somatic vulnerability factor. Factor 3 had salient loadings from all five variables from the acute phase, with negative loadings from GCS and GOS at discharge from the acute hospital, and positive loadings from length of ICU stay, AIS_head and a direct pathway to rehabilitation. This factor was designated as an injury severity factor.

New factor analyses were conducted, one for each factor. Anxiety and depression were cross-loaded on factors 1 and 2 and were selected for inclusion in the psychosocial robustness factor due to stronger loadings. Likewise, the GSCL subscales for gastrointestinal and cardiovascular symptoms were cross-loaded on factors 1 and 2 and were selected for inclusion in the somatic vulnerability factor due to higher loadings and more conceptual overlap. The final factor analyses supported the unidimensionality of the three factors, and the factors demonstrated good to adequate factor reliability. See Table 2 for final factor loadings and reliability indicators.

### 3.5. Multiple Regression

Results from the blockwise multiple linear regression of fatigue in the sample with complete data are shown in Table 3. Age, education and gender had no significant associations with the fatigue factor (Model 1), and the model explains a non-significant amount of variance in fatigue. The injury severity factor did not in isolation contribute significantly to the model in the second regression block.

In Model 3, psychosocial robustness was significantly negatively associated with fatigue, and somatic vulnerability showed a strong positive association with fatigue. The injury severity factor entered in the previous block now showed a barely statistically significant effect. While the effects for the psychosocial robustness factor and the injury severity factor were significant, the confidence intervals bootstrapped for their coefficients border on zero, and as such, demonstrate less robust effects than the somatic vulnerability factor. This final model explains 44.2% of the variance in the fatigue factor.

### 3.6. Post Hoc Analyses

Due to the non-inclusion of the neuropsychological measures in the factors derived from earlier steps, correlations between the residuals of the regression analysis and the neuropsychological measures were inspected. The residual from the final regression model was negatively associated with mental flexibility (CWIT-4, n = 90, r = −0.27) and sustained attention (CPT-III CoV block change, n = 95, r = −0.20). For exploratory purposes, a composite score of these two measures was added in a final block in the blockwise regression (n = 89). The results overlapped considerably with those from the primary regression model. The addition of the neuropsychological composite variable in the final block led to a significant increase in explained variance up to 51.6%. However, the neuropsychological composite score was negatively associated with the injury severity factor (n = 89, r = −0.23), and its inclusion suppressed the association of the injury severity factor below significance (see Appendix A).

Finally, the relative importance of each variable loading upon the three factors was explored in univariate regression models, with the fatigue factor as the dependent variable. For univariate regression coefficients and explained variance, see Appendix A. The anxiety, depression and the resilience subscale, planned future, had the strongest univariate impact on fatigue in the psychosocial robustness factor. In the somatic vulnerability factor, all variables explained a significant amount of variance in fatigue, but the GSCL musculoskeletal symptoms subscale demonstrated the strongest positive association. Finally, for the injury severity factor, effects were in general weak, and only the Direct Pathway to Rehabilitation and AIS_head demonstrated significant univariate associations with fatigue.

## 4. Discussion

The present study aimed to explore dimensions underlying various biopsychosocial constructs commonly associated with fatigue six months following TBI. In line with the notion of fatigue as being influenced by both injury-specific and general risk factors, this study examined the relationship between a multitude of variables that have previously been associated with fatigue after TBI, and several fatigue outcome measures. The results highlight that three underlying factors related to psychosocial robustness, somatic vulnerability and injury severity can be identified, providing a clearer picture of the somewhat fragmented literature on protective and risk factors for post-TBI fatigue.

### 4.1. Unidimensionality of Post-TBI Fatigue

Regarding fatigue levels, our findings confirm variations between measures. On the FSS, the patients reported similar levels of fatigue interference as those seen in the general population [48]. On the CFQ, however, the sample reported fatigue symptoms approximately one standard deviation above the normative average [51], and on specific items, 60% reported increases in tiredness and their need for rest. Our findings support the notion that the majority of patients with TBI experience increased levels of fatigue, while many, despite their symptoms, report little to no interference from fatigue during the first 6 months. This aligns with the findings by Kjeverud et al. [38] in stroke patients, which were interpreted as a dissociation between fatigue severity and fatigue interference. Some patients may experience more fatigue following injury but are able to compensate successfully such that it does not interfere with the roles and activities pertinent to their daily life. Additionally, many patients were still on sick leave at the time of measurement, which could contribute to a low degree of functional interference due to decreased environmental demands.

Despite these variations, the items from the included fatigue PROMS demonstrated good reliability and considerable unidimensionality in our factor analytic approach, indicating that the measures seem to measure a uniform concept. The single fatigue item from the RPQ also demonstrated good correspondence with the other measures, which support the utility of this single item in clinical practice, and items from the GSCL fatigue subscale also aligned well along the unidimensional fatigue factor. Items 1 and 2 from the FSS did not load saliently on the fatigue factor, in line with previous studies of the FSS in patients with, e.g., stroke [75], and were thus not included.

### 4.2. Biopsychosocial Dimensions-Relevance for Fatigue

Through factor analyses, we evaluated overlap and underlying dimensionality among self-reported PROMS of biopsychosocial constructs often associated with fatigue. Two salient factors were extracted, which we termed psychosocial robustness and somatic vulnerability. These factors showed some overlap with regard to anxiety and depression, as well as gastrointestinal and cardiovascular symptoms, showing that there are some commonalities between them despite the parsimonious structure selected. A third factor was found, termed as an injury severity factor based on strong loadings from injury-related severity indices from an acute hospital stay. In the subsequent multivariate regression analyses, somatic vulnerability, psychosocial robustness and injury severity factors all demonstrated significant associations with fatigue, explaining 44.2% of variance in fatigue 6 months after TBI.

Somatic vulnerability demonstrated a particularly strong and robust association with fatigue, in line with the literature linking pain and fatigue as central comorbidities [22,76], and earlier studies in the TBI population [23,25]. This factor explained 39% of the variance in fatigue in isolation, in essence contributing most of the explained variance in the multivariate regression models. Subsequent univariate post hoc regression analyses showed that all the variables underlying this dimension contributed significantly to the association between somatic vulnerability and fatigue. Notably, the GSCL subscale for musculoskeletal symptoms explained more variance in fatigue than the somatic vulnerability factor in large, indicating that nonspecific musculoskeletal pains are particularly crucial markers for somatic vulnerability and the factor’s association with fatigue in this sample.

The association between psychosocial robustness and fatigue supports earlier findings linking resilience with less fatigue after TBI [77]. Trait extraversion, conscientiousness and optimism seemed to align with resilience factors in this protective dimension, while trait neuroticism, loneliness, behavioral inhibition and psychological distress were placed on the opposite side of this dimension, confirming that absence of negative emotionality is a prominent feature of psychosocial robustness. Associations between high neuroticism, low extraversion and low conscientiousness and fatigue have been demonstrated in mild TBI [34] and other populations [78]; thus, these findings are in line with previous findings. While trait extraversion, trait conscientiousness and trait optimism did load heavily on this protective dimension, they were not significantly associated with fatigue 6 months post-injury in isolation. Conversely, measures of state and trait negative affectivity (state depression and anxiety, and trait neuroticism to a lesser degree) and resilience (planned future, and to a lesser degree perception of self) were essential to the relevance of psychosocial robustness for fatigue in our sample. The resilience subscale for planned future pertains to the perception of the future as manageable and predictable through goal-directedness and structure, while the subscale for perception of self relates to self-efficacy and potential for growth through adversity. These constructs thus align well as opposites to anxiety and depression.

The association between fatigue and injury severity became significant when controlling for psychosocial robustness and somatic vulnerability. Among the underlying injury-related variables, only the direct pathway to rehabilitation and the AIS_head demonstrated significant univariate associations with fatigue in post hoc regression analyses, indicating that anatomical brain injury severity combined with early functional status are particularly relevant. Post hoc analyses furthermore demonstrated that a measure of mental flexibility suppressed the association between the injury severity factor and fatigue, indicating that the injury severity factor from the acute phase and the resulting cognitive deficits in mental flexibility after six months overlap in their contributions to fatigue.

A visual representation of the findings is provided in Figure 2.

### 4.3. Implications for Rehabilitation

The fact that fatigue was strongly associated with functional status 6 months post-injury is in line with earlier findings. The results illustrate that fatigue is associated with everyday functioning and point to the importance of addressing fatigue in rehabilitation [2]. While fatigue is a severe problem for many patients with TBI, there is nevertheless considerable heterogeneity, with some patients reporting little to no fatigue interference in everyday life. Understanding which patients are at risk of developing persistent fatigue and functional interference from fatigue, and why, is crucial in improving our care for this patient group.

While more severe injuries are accompanied by greater sensory-motor and cognitive deficits, and accordingly might necessitate greater compensatory efforts in returning to mental and cognitive activities, initial injury severity indices were inconsistently associated with fatigue in our study. Our findings showed that some brain injury severity indices and having a direct pathway to rehabilitation were weakly associated with fatigue. The latter finding may likely be interpreted as a proxy for functional status, as patients with severe symptoms were more likely to be transferred to rehabilitation, irrespective of injury severity measures. The injury severity factor was only associated with fatigue when controlling for robustness and vulnerability, confirming that other risk factors for fatigue are intertwined with injury severity initially, but can be disentangled when adjusted for. For instance, patients with relatively mild injuries, but who suffer from co- or premorbid pain or depression, may be at high risk for fatigue despite mild injuries. While having a high degree of somatic vulnerability and low degree of psychosocial robustness might contribute to an increased risk of fatigue in isolation, injury characteristics serve as an independent risk as well, although these associations are less robust.

Our findings also underline the importance of the contribution of various biopsychosocial protective and vulnerability factors. Somatic symptom burden and especially pain emerge as important associated factors with fatigue following TBI, which should be considered as central targets for rehabilitation. The exact nature of the relationship between fatigue and pain cannot be deduced based on our cross-sectional design, but until further longitudinal research sheds more light on these relationships, the possibility of temporal and bidirectional influences should be considered. Rehabilitation efforts addressing fatigue should therefore also address concurrent risk factors for fatigue. This can be achieved through holistic rehabilitation programs. New methods such as virtual reality have shown promising results in the treatment of pain, emotional symptoms, and fatigue, and should be explored [79,80].

This study furthermore demonstrates the importance of taking into account protective factors which might buffer against fatigue. Aspects of resilience such as perceiving the future as manageable and predictable, and self-efficacy in the face of adversity, were negatively associated with fatigue. On the opposite side of the same dimension, lower levels of loneliness and negative effects are positively associated with fatigue. The findings indicate that rehabilitation efforts aimed at helping patients re-establish a coherent sense of self and their future, and to reconnect with social resources, might lessen their risk of fatigue in the early stages of rehabilitation. This latter point was supported in a recent qualitative study [81], in which the use of social support was identified as a promising treatment angle for breaking vicious cycles for perpetuation and exacerbation of fatigue after brain injury.

### 4.4. Limitations

This study examined cross-sectional associations between fatigue and related constructs but did not allow for inferences regarding directional influences. Furthermore, while dimensions derived from factor analyses provide a parsimonious structure to the relations between various predictors of fatigue, one cannot eliminate possible within- and between-factor dynamics, such as premorbid trait neuroticism influencing the post-injury development of anxiety and depression, which could again influence fatigue. Our post hoc analyses furthermore demonstrated that the variable loading on each factor contributed to different degrees of fatigue when viewed in isolation. Finally, our study has a relatively modest sample size, and generalizations of the results to other cohorts should be made with caution. Of 450 patients with intracranial injury admitted to the Neurosurgery department in the study period, we assessed 55% for eligibility and included 21.3% of the total population. The mean age and the gender ratio included are in line with the TBI population included in the quality database [44]. However, our sample is weighted toward moderate and severe injuries (77%) compared with those included in the quality database (57%). Thus, the results may not be generalizable to those with milder intracranial injuries.

Ideally, a somewhat larger sample would have to be investigated to provide better estimates of essential parameters (particularly factor loadings and regression coefficients) in the population in question. However, while the parameter estimates could be more accurate, and small sample sizes tend to increase the liability to Type II errors, and we see no reason to doubt the general pattern of findings from the study.

## 5. Conclusions

Through the exploration of factors associated with fatigue following TBI, this study has demonstrated that factors related to fatigue after TBI might be described along three dimensions, i.e., psychosocial robustness, somatic vulnerability and injury-related factors. Within these factors, somatic symptom burden (especially pain), depression, anxiety, positive perceived prospects for the future, loneliness daytime sleepiness, subjective insomnia, anatomical severity of injury and being referred directly to rehabilitation services all demonstrated relevance for fatigue 6 months post-injury. These factors, while having varying importance, illustrate the breadth of biopsychosocial underpinnings for fatigue following TBI.

The findings illuminate potential tangible treatment targets in rehabilitation of fatigue after TBI and may guide future research aimed at establishing evidence-based treatment options. More research is needed to understand potential dynamic interactions between fatigue and the associated vulnerability and protective factors, and to understand how these may develop over time.

## Figures and Tables

**Figure 1 jcm-11-01733-f001:**
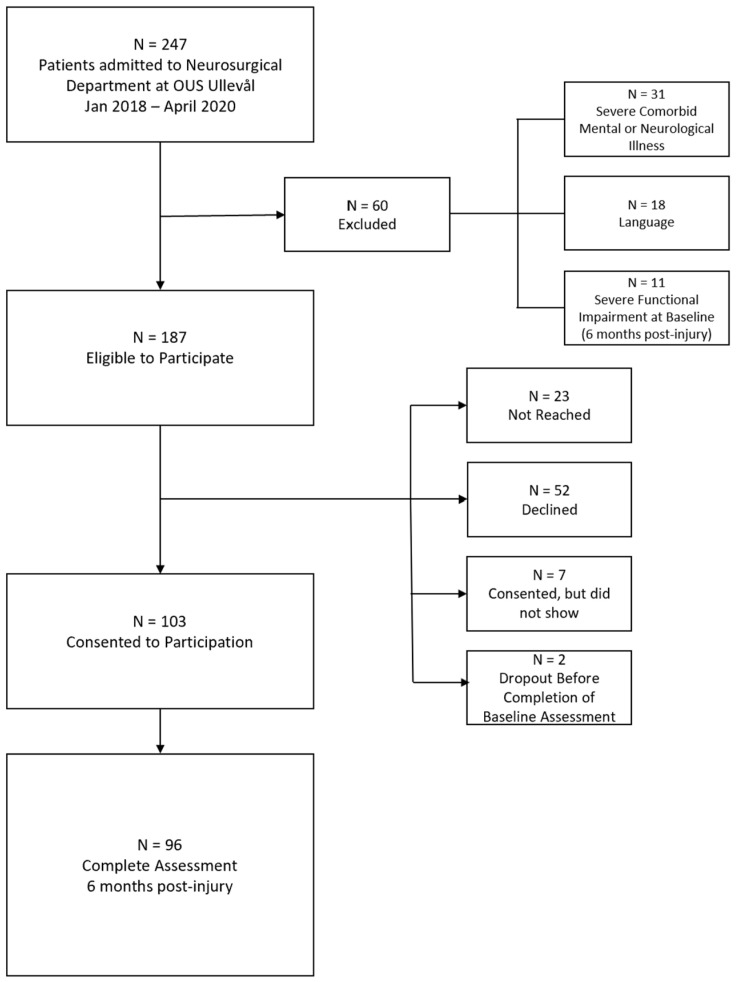
Flow chart of the inclusion and exclusion process. From a sample of 187 eligible patients, 103 participants (55%) consented to participate, and 96 ended up with a complete dataset.

**Figure 2 jcm-11-01733-f002:**
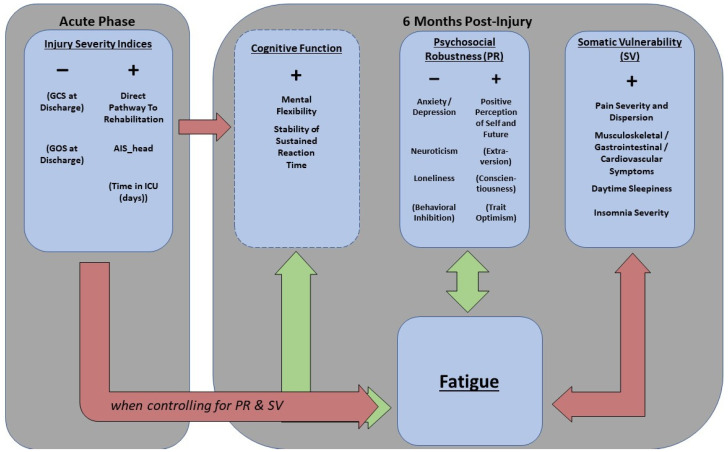
A visual representation of the findings from our study. Note that cognitive function is marked by a dotted box, so as to illustrate that these effects were found in post hoc analyses with a slightly smaller sample due to missing data. Double-sided arrows imply within-time associations, while one-sided arrows imply unidirectional influences. Green arrows imply positive correlations, and red arrows imply negative correlations. Parentheses signify variables with significant loadings on the factor, but with no significant contribution to fatigue when inspected in isolation.

**Table 1 jcm-11-01733-t001:** Factor loadings of items from fatigue measures. All items load saliently on the component at significance level of *p* < 0.05, i.e., loadings above 0.40.

	Fatigue
	Component
FSS Item 3	0.80
FSS Item 4	0.44
FSS Item 5	0.76
FSS Item 6	0.73
FSS Item 7	0.80
FSS Item 8	0.78
FSS Item 9	0.82
CFQ Item 1	0.81
CFQ Item 2	0.70
CFQ Item 4	0.63
CFQ Item 5	0.80
CFQ Item 6	0.56
CFQ Item 7	0.54
GSCL Item 1	0.61
GSCL Item 12	0.85
GSCL Item 17	0.69
RPQ Item 6	0.81
Extraction Sum of Squared Loadings (% of variance)	8.9 (52.4%)
Cronbach’s alpha	0.95

**Table 2 jcm-11-01733-t002:** Factor loadings for the final unidimensional factor analyses of self-reported independent variables (N = 96). Squared loadings and explained variance therefore refer to only those variables included in each of the three factor analyses. For an overview of the primary factor analyses, see the Appendix A.

	Factors		
	Psychosocial Robustness	Somatic Vulnerability	Injury Severity
Behavioral Inhibition	−0.55		
Trait Neuroticism	−0.90		
Trait Extraversion	0.63		
Trait Conscientiousness	0.56		
Trait Optimism	0.69		
Loneliness	−0.70		
Anxiety Symptoms	−0.64		
Depressive Symptoms	−0.76		
Resilience–Perception of Self	0.84		
Resilience–Planned Future	0.64		
Daytime Sleepiness		0.48	
Insomnia Severity Index		0.48	
Pain–Affected Regions		0.74	
Strongest Pain		0.84	
Weakest Pain		0.64	
Average Pain		0.88	
Current Pain		0.73	
Gastrointestinal Symptoms		0.61	
Musculoskeletal Symptoms		0.84	
Cardiovascular Symptoms		0.53	
AIS_head			0.58
Length of ICU Stay (days)			0.58
GCS at Discharge			−0.67
GOS at Discharge			−0.77
Direct Pathway to Rehabilitation			0.71
Extraction Sums of Squared Loadings (% of variance in included variables)	4.9 (49.0%)	4.8 (48.0%)	2.2 (44.4%)
Cronbach’s alpha	0.91	0.89	0.80

**Table 3 jcm-11-01733-t003:** Blockwise multiple linear regression (N = 96). Unstandardized (B) and standardized coefficients (β) are reported. Adjusted R^2^ shows the model-explained variance, and the F change-statistic is a test of the improvement from the previous model. Standard errors (SE) shown are calculated from bootstrapping. The final column shows the 95% confidence interval for the unstandardized coefficients (B) in Model 3. ^ns^ not significant, * *p* < 0.05, *** *p* < 0.001.

	Model 1		Model 2		Model 3		95% CI	
	β	B (SE)	β	B (SE)	β	B (SE)	Lower	Upper
Constant		−0.08 (0.14)		−0.08 (0.11)		−0.08 (0.09)	(−0.25	0.09)
Age (Centered)	0.01	0.00 (0.01)	0.00	0.00 (0.01)	−0.01	−0.00 (0.01)	(−0.01	0.01)
Education (Centered)	0.00	0.00 (0.04)	0.01	0.00 (0.01)	0.10	0.05 (0.04)	(−0.02	0.13)
Female	0.17	0.41 (0.26)	0.17	0.40 (0.27)	0.12	0.29 (0.18)	(−0.08	0.65)
Injury Severity			0.13	0.14 (0.11)	0.16 *	0.18 (0.08)	(0.01	0.34)
Psychosocial Robustness					−0.17 *	−0.17 (0.09)	(−0.34	−0.01)
Somatic Vulnerability					0.59 ***	0.60 (0.08)	(0.46	75)
Adjusted R^2^	0.001		0.001		0.442			
F Change	0.89 ^ns^		1.65 ^ns^		36.8 ***			

## Data Availability

Due to the sensitive nature of the data involved in this project, the data have not been made publicly available. Interested parties may contact the corresponding author (D.L.) for requests for data access.

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
