# Peer review of "Impact of Somatic Vulnerability, Psychosocial Robustness and Injury-Related Factors on Fatigue following Traumatic Brain Injury—A Cross-Sectional Study"

_jcm, 2022, doi:10.3390/jcm11061733_

Round 1

Reviewer 1 Report

Interesting topic, please look at these points:

  • Lines 136-137: "Finally, the relevance of these dimensions to fatigue 6 months after TBI was explored. " Is this point 3) ? Revise.
  • Is 1.2 subsection "Clinical Complexity" technically important? Why did authors decide to separate it?
  • Lines 38-39: "TBI is defined as "an alteration in brain function, or other evidence of brain pathology, caused by an external force" [4]." Improve this concept. TBI causes high mortality and morbidity. Look at these 2 important refs: -- Posttraumatic synchronous double acute epidural hematomas: Two craniotomies, single skin incision. Surg Neurol Int. 2020 Dec 11;11:435. doi: 10.25259/SNI_697_2020. --  Surgical Management of Vertex Epidural Hematoma: Technical Case Report and Literature Review. World Neurosurg. 2017 Jul;103:475-483. doi: 10.1016/j.wneu.2017
  • Lines 559-560: "The fact that fatigue was consistently linked to functional status 6 months post-injury, supports the notion of fatigue as an important target in rehabilitation" What do authors mean in this sentences ?
  • In "4.3. Implications for Rehabilitation" subsection. Authors should discuss about the current role of virtual reality in rehabilitation in neurosurgery and trauma. Look at these refs: -- doi: 10.3233/NRE-172361 and 10.1007/s10439-021-02834-8.
  • Lines 527-529: "Associations between high neuroticism... TBI [30] and other populations [74]" What about your results?
  • Lines 84-86: "Depression and anxiety are also related to fatigue following TBI [22]–[25]. While fatigue may... acquired brain injury [24], suggesting that the two are related, but distinguishable" What about fatigue and anxiety? Add refs.

Reviewer 2 Report

In the present study, the authors aim to "provide a comprehensive exploration of factors relevant to the identification and rehabilitation of fatigue following TBI."

The study is carefully prepared and clearly presented. The topic is very beneficial for practice and the results may help further studies that should bring more concrete to clinical practice.

I consider the only shortcoming to be the small sample that was monitored.

The introductory passage is quite extensive. Given that the authors also set the goal "to enhance our theoretical understanding of the relationship between fatigue and injury-related, cognitive and self-reported biopsychosocial factors", the introduction can be accepted.

The methodological procedures are written systematically, all used procedures are adequate and correspond to the objectives of the study. The only thing that is not entirely clear from the description is whether the patients included in the study were only those who were registered at Oslo University. If this is the case, it would be appropriate to provide information on how many patients with TBI were in Norway in the observed years or what percentage of the 96 patients enrolled in the study accounted for the total number of TBI patients in Norway.

The discussion is extensive and is mainly devoted to the issue of fatigue. However, it was appropriate to include in the discussion considerations about the possible influence of the sample size of the monitored patients on the results.

Round 2

Reviewer 1 Report

Good